

# Response of the link between ENSO and the East Asian winter monsoon to Asian anthropogenic aerosols

Zixuan Jia[1], Massimo A Bollasina[2], Wenjun Zhang[1], Ying Xiang[3]

[1]School of Atmospheric Science, Nanjing University of Information Science and Technology, Nanjing, China
[2]School of GeoSciences, University of Edinburgh, Edinburgh, UK
[3]Jiangsu Climate Center, Nanjing, China

*Correspondence to*: Zixuan Jia (zx.jia@nuist.edu.cn)

**Abstract.** We use coupled and atmosphere-only simulations from the Precipitation Driver and Response Model Intercomparison Project to investigate the impacts of Asian anthropogenic sulfate aerosols on the link between the El Niño-Southern Oscillation (ENSO) and the East Asian Winter monsoon (EAWM). In fully-coupled simulations, aerosol-induced cooling extends southeastward to the Maritime Continent and the north-western Pacific. Remotely, this broad cooling weakens the easterly trade winds over the central Pacific, which reduces the east-west equatorial Pacific sea surface temperature gradient. These changes contribute to increasing ENSO's amplitude by 17%, mainly through strengthening the zonal wind forcing. Concurrently, the El Niño-related warm SST anomalies and the ensuing Pacific-East Asia teleconnection pattern (i.e. the ENSO-EAWM link) intensify, leading to an increased EAWM amplitude by 18% in the coupled simulations. Therefore, in response to the increasing frequency of El Niño and La Niña years under Asian aerosol forcing, the interannual variability of the EAWM increases, with more extreme EAWM years. The opposite variations in the interannual variability of the EAWM to Asian aerosols in atmosphere-only simulations (-19%) further reflect the importance of ENSO-related atmosphere-ocean coupled processes. A better understanding of the changes of the year-to-year variability of the EAWM in response to aerosol forcing is critical to reducing uncertainties in future projections of variability of regional extremes, such as cold surges and flooding, which can cause large social and economic impacts on densely populated East Asia.

## 1 Introduction

The East Asian winter monsoon (EAWM) is one of the most prominent features of the northern hemisphere atmospheric circulation during the boreal winter, and has a pronounced influence on weather and climate of the Asian-Pacific region from the northern latitudes to the equator (e.g. Chang, 2006). As such, variations of the EAWM have the potential to cause extreme cold disasters and severe flooding in Southeast Asian countries (e.g. Feng et al. 2010; Huang et al. 2012), with consequent marked social and economic impacts (e.g. Chen et al., 2005; Zhou et al. 2011). Thus, it is very important to



understand the mechanisms underpinning its variability and associated drivers, and to ultimately
develop more robust projections of its future evolution.
The EAWM is fundamentally driven by the thermal contrast between the cold Asian continent and the
adjacent warm oceans (e.g. Yang et al., 2002; Huang et al., 2012; Chen et al., 2019). Its climatological
pattern is mainly characterized by dry cold low-level northwesterlies along the eastern flank of the
Siberian High and low-level northeasterlies along the coast of East Asia, triggering cold air outbreaks
in northern China and generating cold surges over southern China as well as the South China Sea (Li
and Wang, 2012; He et al., 2013). The EAWM exhibits distinct interannual variability (e.g. Gong et al.,
2014), which is strongly influenced by the El Niño-Southern Oscillation (ENSO) and the ensuing
Pacific-East Asia (PEA) teleconnection pattern (e.g. Zhang et al., 1996). Associated with an El Niño
event, the anomalous anticyclone over the western tropical Pacific (the most remarkable low-level
circulation feature of the PEA) induces southwesterlies on its western flank, which weaken the EAWM
flow and lead to warmer and wetter conditions over southeastern China and the South China Sea (Wang
et al., 2000, 2013). In turn, the EAWM tends to be strong during La Niña winters, with widespread
cooling and reduced precipitation.
Previous studies indicated that magnitude and location of ENSO-induced teleconnection patterns are
influenced by ENSO characteristics, such as amplitude and location of its sea surface temperature (SST)
anomalies (Cai et al., 2021; Jiang et al., 2022). However, future projections of ENSO characteristics are
highly uncertain, even in the latest CMIP6 models (Huang and Xie, 2015; Yan et al., 2020; Beobide-
Arsuaga et al., 2021). Therefore, there is no consensus on future changes in the ENSO-induced
teleconnections, including projections of the PEA pattern (e.g. Wang et al., 2013; Jia et al., 2020). The
characteristics of ENSO and its induced atmospheric teleconnections are closely related to the tropical
Pacific mean state via ocean-atmosphere feedbacks (Jin, 1997; Wang, 2002; Cai et al., 2014). Based on
ocean–atmosphere reanalyses, observed mean state changes since the 1980s feature a La Niña-like
warming (i.e. the tropical Pacific warming center is mainly located in the western basin; Rayner et al.,
2003; Kobayashi et al., 2015; Huang et al., 2017). However, both a La Niña-like and an El Niño-like
warming (i.e. tropical Pacific warming centered in the eastern basin) are projected in the future, with a
large spread across different climate models (e.g. Power et al., 2013; Lian et al., 2018). These two
different warming patterns will cause a corresponding strengthening and weakening of the easterly trade
winds over the tropical Pacific Ocean, respectively, resulting in opposite changes in the characteristics
of ENSO (Vecchi et al., 2006; Collins et al., 2010). While the majority of the studies have focused on
the influence of increasing greenhouse gas concentrations on the tropical Pacific mean state (e.g. Wang
et al., 2017; Yan et al., 2020), the impact of anthropogenic aerosols has been largely overlooked.



Due to the intensification of human industrial activities, the global mean atmospheric burden of
anthropogenic aerosols has continued to increase over the past century, exerting a significant imprint
on worldwide climate (Liao et al., 2015; Forster et al., 2021; Persad, 2023). Anthropogenic aerosols can
affect climate by modulating shortwave radiation and, to some extent, longwave radiation directly, and
through their interactions with clouds and precipitation indirectly (Boucher et al., 2013; Myhre et al.,
2013; Zhao and Suzuki, 2019). Unlike greenhouse gases, which are distributed evenly across the globe,
anthropogenic aerosols reside in the atmosphere for a short time (days to weeks) due to numerous
chemical and physical removal processes, which causes their distribution and associated radiative
forcing to be spatially heterogeneous (Allen et al., 2015; Wilcox et al., 2019). As such, aerosols can
induce substantial changes in local atmospheric circulation and extend their influence over long
distances, even over the surrounding ocean, triggering ocean–atmosphere interactions (Rotstayn and
Lohmann, 2002; Ramanathan et al., 2005; Westervelt et al., 2020). Some studies indicated that the
influence of anthropogenic aerosols from remote sources can even outweigh that of locally-emitted ones
(Shindell et al., 2012; Lewinschal et al., 2013). Since the start of the industrial age, vast emissions of
aerosols and their precursors over the Northern Hemisphere have had a profound cooling effect, and
this preferential cooling has been linked to a southward shift of the Intertropical Convergence Zone (e.g.
Hwang et al., 2013; Navarro et al., 2017).
The emissions of anthropogenic aerosols and their precursors in Asia have increased rapidly since 1980,
and many studies have focused on Asian as well as Northern Hemispheric climate (e.g. Bollasina et al.,
2014; Bartlett et al., 2018; Wilcox et al., 2019; Li et al., 2022). While Asian anthropogenic aerosols can
significantly affect the Asian monsoon, the large majority of the current literature has focused on the
effects of aerosols on the summer or annual mean climatology (e.g., Westervelt et al., 2018; Song et al.,
2014; Persad et al., 2022). Only a limited number of studies have focused on the influence of aerosols
on the EAWM (Jiang et al., 2017; Liu et al., 2019; Wilcox et al., 2019), while their effect on the
interannual variability of the EAWM and the link to ENSO remains unexplored. In boreal winter, coal
and fossil fuels are combusted intensively across Asia (Gao et al., 2018; Cheng et al., 2019), setting the
stage for a potential important influence on continental climate and the mean EAWM circulation.
Moreover, ENSO and the associated PEA teleconnection pattern peak in the winter, representing a
major driver of interannual fluctuations of the EAWM. The extent to which aerosols may affect ENSO
and the related ocean-atmosphere feedbacks has not been thoroughly investigated and is unclear
(Westervelt et al., 2018; Wilcox et al., 2019). Given the rapid variations in aerosol emissions over Asia,
addressing this knowledge gap is both compelling and timely for enhancing our understanding and
projections of the ENSO-EAWM link in the near future, and potential causes of changes in the
interannual variability of the EAWM.



In this study, we use multi-model mean data from regional aerosol perturbation experiments conducted
with coupled and atmosphere-only models (Section 2) to investigate the impacts of Asian aerosols on
the ENSO-EAWM link and the interannual variability of the EAWM (Section 3). We then link changes
in the PEA pattern to the remote impacts of Asian aerosols on ENSO (Section 4). Mechanisms driving
changes in the tropical Pacific mean state and ENSO characteristics are further investigated in Section
5. Finally, Section 6 summarises the main results and provides key conclusions.
**2 Data and methodology**
Model data from the Precipitation Driver and Response Model Intercomparison Project (PDRMIP;
Myhre et al., 2017) are used to investigate the impact of Asian anthropogenic aerosols on the ENSO-
EAWM link. PDRMIP offers a unique opportunity for elucidating the complexities of the slow and fast
responses of the EAWM to Asian aerosols and the contribution of ENSO-related ocean-atmosphere
coupled processes with coupled and atmosphere-only simulations by comparing baseline and regional
aerosol perturbation experiments. The baseline simulation was forced by present-day (year 2000) levels
of aerosols and greenhouse gas emissions/concentrations. The regional aerosol experiment analysed in
this study has sulfate concentrations/emissions over Asia (10°–50°N, 60°–140°E) increased by a factor
of 10 compared to the baseline values (hereafter SUL×10Asia). Note that sulfate is the predominant
aerosol component in boreal winter over Asia (e.g. Liu et al., 2009; Zhang et al., 2018). The response
to Asian aerosols is identified as the difference between the SUL×10Asia and the baseline experiments.
Of the 10 models that contributed to PDRMIP, seven performed the SUL×10Asia experiment: GISS-
E2, HadGEM3-GA4, IPSL-CM5A, MIROC-SPRINTARS, ESM1-CAM4, CESM1-CAM5, and
NorESM1 (details on the resolution and aerosol setup for each model can be found in Table 1 of Liu et
al. (2018)). For each model and experiment, a pair of simulations was performed: one in a fully coupled
atmosphere–ocean setting (called "coupled"), and one with fixed climatological sea surface
temperatures (called fSST). The coupled simulations were run for 100 years and the fSST simulations
for 15 years. The concentrations of all non-aerosol anthropogenic forcers and natural forcing were kept
at present-day levels (typically year 2000) in all the experiments, as are the SSTs for the fSST
simulations. In this study, we use output from the last 50 winters (DJF, December of the current year
and January and February of the following year) of coupled simulations and the last 12 winters of the
fSST simulations to discard the model spin-up time and consistently with existing literature (Liu et al.,
2018; Dow et al., 2021; Fahrenbach et al., 2024).

Reanalysis and observational data for DJF 1965–2014 (50 years) are used to evaluate the PDRMIP-
simulated EAWM and ENSO-related patterns in the baseline experiment. Monthly meteorological
reanalysis data are from the fifth-generation atmospheric reanalysis ERA5 provided by the European
Centre for Medium-Range Weather Forecasts at a spatial resolution of 0.25° (Copernicus Climate





Change Service, 2017; Hersbach et al., 2023). Monthly gridded observations are from the Hadley Centre
Sea Ice and Sea Surface Temperature (HadISST) dataset for sea surface temperature at a spatial
resolution of 1° (Rayner et al., 2003), and from the Climatic Research Unit (CRU) v4.07 data set for
land surface temperature with a spatial resolution of 0.5° (Harris et al., 2020). To quantify the EAWM
interannual variability, we use the Ji et al. (1997) index (the negative 1000 hPa meridional wind
anomaly averaged over 10°–30°N, 115°–130°E) as it represents the spatio-temporal characteristics of
the ENSO–EAWM relationship well (Gong et al., 2015; Jia et al., 2020). Positive values indicate a
stronger-than-normal EAWM. ENSO is described by the Niño3.4 index (area-averaged SST anomaly
over 5°S–5°N, 120°–170° W). The ENSO-related PEA pattern is deduced by regression analysis, and
the statistical significance is evaluated using the two-tailed Student's *t-test*. Among the seven PDRMIP
models with the SUL×10Asia experiment, coupled baseline simulations in CESM1-CAM5, MIROC-
SPRINTARS, HadGEM3-GA4, and NorESM1 can well capture the observed pattern and magnitude of
the ENSO-related circulation anomalies across East Asia and the Pacific (Fig. S1) and are used in this
study. These four models include parameterisations of both aerosol-radiation and aerosol-cloud
interactions, while the others don't include indirect effects, or include only the first indirect effect (Liu
et al., 2018; Dow et al., 2021). (Table 1). All the data are interpolated to a 3.75° × 2° (longitude ×
latitude) resolution before the analysis for consistency between all models.
**3 Impacts of Asian aerosols on the PEA pattern and the EAWM interannual variability**
The ENSO-related circulation and precipitation anomalies across East Asia and the Pacific (i.e. the PEA
pattern) (Figs. 1a-c) are well reproduced by the multi-model mean of the PDRMIP coupled baseline
simulations (Figs. 1d-f). The pattern is characterised by the El Niño-related warm SST anomalies over
the equatorial Pacific and cold SSTs over the north-western Pacific (Fig. 1a), the anomalous anticyclone
over the western tropical Pacific and the anomalous low over the northern extratropical Pacific (Fig.
1b). On the western flank of the anticyclone, near-surface and lower tropospheric southerly winds along
the East Asian coast (Figs. 1a-b) lead to warm surface air temperature and precipitation over
southeastern China and even over central China (Figs. 1a, c), while the lower tropospheric northerly
winds on the western flank of the cyclone bring cold air to northeastern China (Fig. 1b). The spatial
patterns of simulated anomalies are broadly similar to those found in observations, including the
position and magnitude of El Niño-related warm SST anomalies, anticyclone and cyclone anomalies,
and precipitation anomalies (Figs. 1d-f). The multi-model mean from PDRMIP shares common biases
with other CMIP5 and CMIP6 models, such as a slightly westward shift of the equatorial Niño warming
with associated circulation and precipitation anomalies (Gong et al., 2015; Wang et al., 2022). Overall,
the multi-model mean coupled PDRMIP baseline simulations successfully reproduce the PEA pattern.



In response to Asian aerosols, the El Niño-related warm SST anomalies intensify over the eastern
equatorial Pacific, associated with an intensification of the anomalous SST cooling over the western
tropical Pacific (Figs. 1d, g). Concurrently, the anticyclonic anomalies over the western tropical Pacific
strengthen and stretch northwestward, while the cyclone over the northern Pacific strengthens and
covers a broader region (Figs. 1e, h). This enhanced anticyclone results in an intensification of southerly
anomalies along the Asian coast from the South China Sea (Figs. 1g-h), advecting warm and humid air
(Figs. 1f, i). Over land, warm and wet anomalies over southeastern and central China weaken, as well
as cold anomalies over northeastern China (Figs. 1d, f, g, i), consistent with the changes in the
atmospheric circulation patterns mentioned above. Overall, these changes suggest that the ENSO signal
and its induced PEA pattern enhance under increased Asian aerosols. Given the interannual variability
of the EAWM is strongly influenced by the PEA pattern, the intensification of southerly anomalies
along the Asian coast associated with the enhanced PEA may lead to an increase in the interannual
variability of the EAWM.
Changes in the interannual variability of the EAWM in response to Asian aerosol increase are shown
by the probability distributions of the EAWM index (Fig. 2). The simulated amplitude of the EAWM
(defined as the standard deviation of the EAWM index) is smaller than the observed amplitude in
baseline simulations, which is a general known bias in models (Wang et al., 2010; Gong et al., 2014).
In coupled simulations, the multi-model mean EAWM amplitudes are 0.55 and 0.65 m s$^{-1}$ for the
baseline and SUL×10Asia experiments, respectively, indicating an 18% increase due to the Asian
aerosols, together with more extreme EAWM years in the SUL×10Asia experiment (Fig. 2a). These
changes are consistent with the aerosol-enhanced PEA pattern identified above. However, in fSST
simulations, the multi-model mean EAWM amplitude decreases by 19%, accompanied by more strong-
EAWM years and less weak-EAWM years in SUL×10Asia experiments (Fig. 2b). These changes can
be explained by aerosol-induced cooling over the emission region and the formation of an anomalous
anticyclonic circulation (e.g. Hu et al., 2015; Liu et al., 2019; Dow et al., 2021), and indicate an
enhanced climatological pattern of the EAWM under increased aerosols (Figs. S2a-f).  In addition to
this atmospheric-only response, the influence of Asian aerosols can extend over the Maritime Continent
and the north-western Pacific (Wilcox et al., 2019; Dow et al., 2021). In coupled simulations, the
climatological pattern of the EAWM extends southeasterly, which is mainly represented by an
anomalous anticyclone centred over the southwest of the Philippines (Figs. S2g-i). This anomalous
anticyclone, attributed to the southward shift of the Hadley circulation to compensate for the
interhemispheric asymmetry in aerosol radiative cooling (Liu et al., 2019), enhances the northerlies
over the Maritime Continent but slightly weakens the northerlies along the East Asian coast (Figs. S2g-
h). This pattern cannot explain the increased interannual variability of the EAWM in coupled
simulations as it is not associated with an evident modulation of the climatological monsoon flow.  The
EAWM-related circulation and precipitation anomalies brought about by increased aerosols in the



coupled experiments (Fig. S3) feature an enhanced PEA pattern. This further suggests the contribution
of the enhanced ENSO-induced PEA pattern to increased interannual variability of the EAWM. The
opposite variations in the interannual variability of the EAWM to Asian aerosols in fully coupled
experiments and atmosphere-only (+18% and -19%, respectively) also reflect the importance of ENSO-
related atmosphere-ocean coupled processes.
**4 The response of ENSO amplitude to increased Asian aerosols**
Following previous studies (e.g. Wang et al., 2013; Wang et al., 2022), the increased ENSO signal and
its induced teleconnection pattern can be further linked to changes in the ENSO amplitude (defined as
the standard deviation of the Niño3.4 index). Figure 3a shows the observed standard deviation of SST
across the tropical Pacific, with the highest values over the central-eastern equatorial Pacific. This
spatial pattern is well captured by the multi-model mean in the coupled baseline simulation (Fig. 3b),
albeit the core values are slightly underestimated in magnitude and spatial extent, especially in the
meridional direction. Increased aerosols lead to significant increases in the SST standard deviation over
the Maritime Continent and the central-eastern equatorial Pacific (Fig. 3c). This is consistent with the
increased ENSO signal and the related changes in SST anomalies over these two regions (Fig. 1g).
Figure 3d shows the probability distributions of the Niño3.4 index from the coupled baseline (blue curve
and shading) and SUL×10Asia (red curve and shading) simulations. The multi-model mean ENSO
amplitude increases by 17% under aerosol forcing (from 0.7 °C to 0.82 °C).
Consistently with the increased ENSO amplitude, Table 1 shows that there are more El Niño (Niño3.4
index > 0.5 ℃) and La Niña (the Niño3.4 index < -0.5 ℃) years in the coupled SUL×10Asia simulation
compared to the baseline for each model, with the increase up to 100% (from 14 to 28 events in the 50-
year record). Figure 4 shows the joint distributions of multi-model mean aerosol-driven changes in the
Niño3.4 index compared with the EAWM index in coupled simulations. Both the Niño3.4 index and
the EAWM index have a wide range of variations (i.e. from -1.5 to 1.5 ℃ and -1 to +1 m s$^{-1}$ respectively),
suggesting that both the ENSO amplitude and the interannual variability of the EAWM increase under
Asian aerosol forcing as indicated above. Remarkably, changes in the Niño3.4 index are significantly
anti-correlated ($p < 0.01$) with those in the EAWM index ($r = -0.38$). In particular, when the Niño3.4
index decreases by less than 0.5 ℃ due to Asian aerosol forcing, the EAWM is 2.5 times more likely to
strengthen than weaken, and vice versa. This is consistent with the negative relationship between ENSO
and the EAWM induced by the ensuing PEA teleconnection pattern (Wang et al., 2000). These results
show that Asian aerosols lead to an increase in the ENSO amplitude, resulting in increased interannual
variability of the EAWM through the associated PEA pattern.



**5 Changes in the tropical Pacific mean state and ocean-atmosphere feedbacks**

It is well-known that ENSO is fundamentally governed by ocean-atmosphere coupled processes in the tropical Pacific (Timmermann et al., 2018; Rashid et al., 2022). It is therefore interesting to examine how the tropical Pacific mean state and atmosphere-ocean coupling are affected by Asian aerosol forcing. Figure 5 shows the climatological annual variation of key surface variables across the equatorial Pacific Ocean in the coupled baseline simulation and their changes under increased Asian aerosols. In the baseline simulation, the equatorial Pacific mean state is characterised by easterly trade winds with maximum magnitude over the central-eastern Pacific, an east-west SST gradient, and strong SST amplitudes (i.e. standard deviations of SST) over the eastern Pacific (Figs. 5a-c). These features are altered in the SUL×10Asia experiment relative to the baseline experiment, with significant seasonal differences. In particular, anomalous westerlies develop from spring over the eastern Pacific, then gradually strengthen until the peak in September while moving towards the central Pacific (the Niño4 region, purple bar) (Fig. 5d). Westerly wind anomalies are considered to play an important role during the development stage (i.e. boreal autumn) of ENSO events, by generating warm SST anomalies in the eastern equatorial Pacific via the thermocline and the advective feedbacks (McPhaden, 1999; Lian and Chen, 2021; Xuan et al., 2024). This anomalous westerly flow weakens the climatological easterly trade winds in the coupled SUL×10Asia simulation compared to the baseline (Figs. 5a, d). Furthermore, anomalous SST warming appears over the eastern Pacific (the Niño3 region, green bar) from autumn to winter (peak around October) (Fig. 5e), which decreases the east-west equatorial Pacific SST gradient (Fig. 5b). Note that Figures 5b and 5e show SST minus zonal mean and SST difference minus zonal mean respectively to clarify the east-west SST changes gradient. Given the broad aerosol-induced cooling over the Pacific (Fig. S2h), warming SST anomalies on Figure 5e represent less cooling. Correspondingly, the SST amplitude increases with maximum values in the winter mainly over the central-eastern Pacific (the Niño3.4 region) (Fig. 5f), which is consistent with the increased ENSO amplitude under Asian aerosol forcing indicated above. Previous studies have found a link between warmer SST in the eastern than in the western equatorial Pacific with an increase in ENSO amplitude (Zheng et al., 2016; Ying et al., 2019; Hayashi et al., 2020).

Given the above marked changes over the equatorial Pacific mean state occur in autumn and winter, we further explore the response of the tropical Pacific mean state to Asian aerosols in these two seasons. In autumn (SON, September-October-November), there are a zonally wider anticyclone, cooling and negative precipitation anomalies stretching from Asia to the whole North Pacific (Figs. 6a-c) compared to those in winter (Figs. 6d-f). As in Figure 5e, Figure 6b and 6c show surface air temperature (SST over the ocean) difference minus domain mean, on which warming SST anomalies represent less cooling. These differences between SON and DJF are related to the climatological pattern in SON when the Siberian High is close to the broad North Pacific subtropical high and the Aleutian Low is weak





(Fig. S4), that lead to the zonally wider cooling by Asian aerosols. The cooling and associated
anticyclonic anomalies trigger cross-equatorial wind anomalies from the Northern Hemisphere to the
Southern Hemisphere, which shift the ITCZ southward (Figs. 6a-c), as indicated by previous studies on
the interhemispheric difference in aerosol emissions (Navarro et al., 2017; Voigt et al., 2017; Wilcox et
al., 2019). Deflected by the Coriolis force, the cross-equatorial wind anomalies present a westerly
anomaly near the equator mainly over the central Pacific (purple box in Fig. 6a), which can weaken the
easterly trade winds, generating warm SST anomalies over the eastern Pacific (green box in Fig. 6b)
and excess rainfall (Fig. 6c). From SON to DJF, the climatological Siberian High strengthens, and the
Aleutian Low deepens with a southward shift in the coupled baseline simulation (Figs. S4a, S2a).
Therefore, the Asian aerosol-induced cooling and associated anticyclone are more concentrated over
the Maritime Continent and the north-western Pacific (Fig. 6d), altering the SST gradient anomaly from
north-south (Fig. 6b) to northwest-southeast (Fig. 6e). This SST anomaly pattern leads to the southward
shift of anomalous westerly winds over the central-eastern Pacific, as well as warm SST and positive
precipitation anomalies over eastern Pacific (Figs. 6d-f). These anomalies are conducive to increasing
the ENSO amplitude.
The processes that most significantly contribute to ENSO are surface wind responses to the equatorial
eastern Pacific SST variations (the Bjerknes or zonal wind feedback), the zonal advection of mean SSTs
by the anomalous current (the zonal advective feedback) and the vertical advection of anomalous
subsurface temperatures by the mean upwelling (the thermocline feedback). The two latter feedbacks
are related to the ocean dynamic responses to zonal wind forcing that cause in-phase variations of
eastern Pacific SST anomalies (Jin and An, 1999; Kim et al., 2014). A diagnostic quantity that includes
both these two feedback processes is the zonal wind forcing of SST anomalies, which was found to be
useful for studying ENSO-amplitude changes under global warming (Rashid et al., 2016). To further
quantify the changes in the strength of the ocean-atmosphere coupling that modulate the ENSO
amplitude, we focus on two main processes, the Bjerknes feedback and zonal wind forcing, which are
related to the formation of the westerly anomalies over the central Pacific and warm SST anomalies
over the eastern Pacific indicated above. Figure 7 shows the lag-regression coefficients between the
SST anomalies averaged over the Niño3 region (green box in Fig. 6b) (the Niño3 SST index) and near-
surface zonal winds (U1000) anomalies averaged over the Niño4 region (purple box in Fig. 6a) (the
Niño4 U1000 index) to represent the Bjerknes feedback and zonal wind forcing. In each panel,
regression coefficients between two variables at different lags are plotted for observations (black curve)
and the coupled baseline (blue curve and shading) and SUL×10Asia (red curve and shading)
simulations. The left panel shows the Niño4 U1000 anomalies response to the Niño3 SST index (i.e.
the Bjerknes feedback). As in most CMIP models (e.g. Bellenger et al., 2014, Rashid et al., 2016), the
simulated Bjerknes feedback is weaker than in observations (Fig. 7a). The strength of the feedback for
lags between −5 and 5 months almost doesn't change in the coupled SUL×10Asia simulation relative





to the baseline (Fig. 7a). The right panel shows the Niño3 SST anomalies response to the Niño4 U1000
index (i.e. the zonal wind forcing). In this case, the simulated SST responses are somewhat stronger
than the observed responses, and the maximum responses are found at small positive lags (e.g. when
U1000 leads SST by 1–2 months) (Rashid et al., 2022). The zonal wind forcing, defined as the
maximum of the regression coefficients (lag=1), strengthens from the baseline (0.51°C m$^{-1}$ s) to the
SUL×10Asia experiment (0.55°C m$^{-1}$ s) by 8%. Therefore, the zonal wind forcing plays a more
important role than the Bjerknes feedback in increasing the ENSO amplitude under Asian aerosol
forcing. In summary, the Asian aerosol-induced cooling weakens the easterly trade winds over the
central Pacific, which reduces the east-west equatorial Pacific SST gradient through the zonal wind
forcing, leading to increased ENSO amplitude.
**6 Summary and conclusions**
This study investigates the response of the ENSO-EAWM link and related interannual variability of the
EAWM to Asian aerosols, including the induced changes in the ENSO-related ocean-atmosphere
feedbacks, using a set of experiments carried out as part of the PDRMIP initiative. Accounting for two-
way atmosphere-ocean coupling, the El Niño-related warm SST anomalies intensify over the eastern
equatorial Pacific, associated with an enhancement of the anomalous anticyclone anomaly over the
western tropical Pacific and corresponding stronger southerlies along the Asian coast from the South
China Sea. This enhanced ENSO signal and its induced PEA pattern contribute to explaining the
increased interannual variability of the EAWM (+18%). When the ocean is not allowed to respond, the
interannual variability of the EAWM varies in the opposite direction (-19%), which further reflects the
importance of ENSO-related atmosphere-ocean coupled processes for explaining the increased
variability. The PEA-like EAWM-related circulation and precipitation anomalies also hint at a link
between increased interannual variability of the EAWM and changes in ENSO in response to Asian
aerosols. The increased ENSO signal can be further linked to changes in the ENSO amplitude. The
multi-model mean ENSO amplitude increases by 17% with increased sulfate aerosols, with more El
Niño and La Niña years in all the PDRMIP models used in this study. Changes in the Niño3.4 index are
significantly correlated with changes in the EAWM index.
In coupled simulations, the aerosol-induced broad cooling alters the mean state over the tropical and
equatorial Pacific, generating westerly anomalies over the central Pacific (peak in autumn) and warm
SST anomalies over the eastern Pacific from autumn to winter, which are key factors in increasing
ENSO amplitude. Using a diagnostic analysis, the contribution of two main processes, the Bjerknes
feedback and zonal wind forcing is estimated. The zonal wind forcing is identified to strengthen from
the baseline experiment to the SUL×10Asia experiment by 8%, while the strength of the Bjerknes
feedback almost doesn't change. Therefore, the aerosol-induced cooling weakens the easterly trade
winds over the central Pacific, which reduce the east-west equatorial Pacific SST gradient through the





zonal wind forcing, causing the increased amplitude of ENSO and the EAWM. In summary, the findings
of this study provide a better understanding of the change to the year-to-year variability of the EAWM
in response to aerosol forcing. This is critical to reducing uncertainties in future projections of
variability of regional extremes, such as cold surges and flooding, which can cause large social and
economic impacts on densely populated East Asia.
We acknowledge some limitations and potential extensions of this study. Only a limited number of
models is available as part of PDRMIP, as some others do not parameterise aerosol-cloud interactions
which are critical to realise the total aerosol response across Asia (e.g. Dong et al., 2016; Liu et al.,
2024). Also, some models prescribed concentrations, rather than emissions, perturbations, the
implications of which are difficult to ascertain given the limited model sample. Including more models
and making use of coordinated perturbed aerosol experiments to Asian aerosols, such as those planned
as part of RAMIP (Wilcox et al., 2023) would further increase the robustness of our study. This would
allow to better characterise the individual model responses as a function of the underlying bias (e.g.,
Liu et al., 2024). It would be interesting to extend this analysis to future projections for the 21[st] century,
for example using CMIP6 models or large ensembles, and examine the externally-forced changes
accounting also for the role of internal climate variability. It would also be interesting to examine the
extent to which the ENSO-EAWM link varies across the various future aerosol pathways, which are
uncertain and display very different, but equally plausible, patterns over Asia (Persad et al., 2022; Wang
et al., 2023). Finally, we only considered the role of Asian aerosol changes. A more comprehensive
analysis, should similar experiments be available, could also consider aerosols from other geographical
regions, such as Europe and North America, which can also affect the Pacific and, via atmospheric
teleconnections, East Asia (e.g. Dong et al., 2016; Liu et al., 2019).
**Code availability.** The python code generated in this study is available upon request (contact author).
**Data availability.** The CRU land temperature dataset is obtained from
https://crudata.uea.ac.uk/cru/data/hrg/cru_ts_4.07, while the HadISST sea surface temperature dataset
can be found at https://www.metoffice.gov.uk/hadobs/hadisst/. The ERA5 reanalysis is provided by the
European Centre for Medium-Range Weather Forecasts at
https://www.ecmwf.int/en/forecasts/dataset/ecmwf-reanalysis-v5. The PDRMIP data can be accessed
through the World Data Center for Climate (WDCC) data server at
https://doi.org/10.26050/WDCC/PDRMIP_2012-2021.

**Author contribution.** ZJ and MAB designed the study and discussed the results. ZJ carried out the
analysis and drafted the manuscript. All authors edited the paper.



**Competing interests.** The authors have no competing interests to declare.
**Acknowledgements.** ZJ thanks the Startup Foundation for Introducing Talent of Nanjing University of
Information Science and Technology (NUIST) (grant no. 2024r034) and Natural Science Fund for
Colleges and Universities in Jiangsu Province (grant no. 24KJB170015). MB acknowledges support
from the Natural Environment Research Council (grant no. NE/N006038/1) and the Research Council
of Norway (grant no. 324182; CATHY).

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



**Figures**

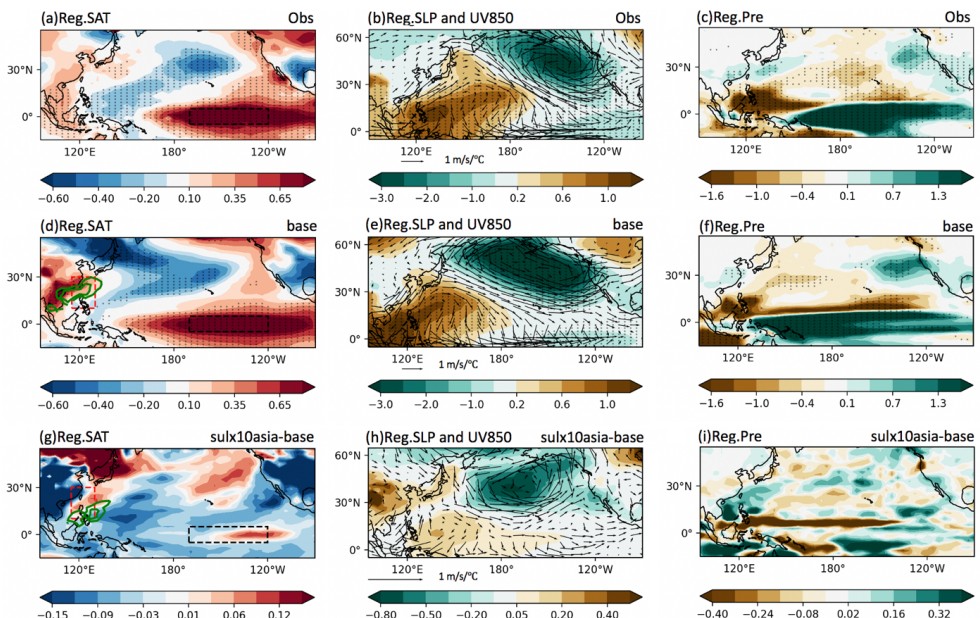

**Figure 1.** DJF regressions of (a)(d) surface air temperature (SAT, SST over the ocean, °C, shading) and 1000 hPa
meridional wind (V1000) over the broad East Asia (green contours, values plotted only when larger than 0.1 m
s$^{-1}$ °C$^{-1}$), (b)(e) sea level pressure (SLP; hPa, shading) and 850 hPa wind (UV850; m s$^{-1}$, vector), (c)(f)
precipitation (Pre, mm d$^{-1}$) onto the Niño3.4 index from coupled (a-c) observations during 1965-2014, (d-f)
multimodel mean coupled baseline simulations in PDRMIP. Dotted regions indicate significant correlations at the
95% level from the two-tailed Student's *t* test. Differences in regressions of (g) SAT and V1000 (green contours,
values plotted only when larger than 0.05 m s$^{-1}$ °C$^{-1}$), (h) SLP and UV850, (i) Pre between coupled SUL×10Asia
and baseline simulations. The definition regions of the EAWM index and the Niño3.4 index are marked by red
and black rectangles respectively.





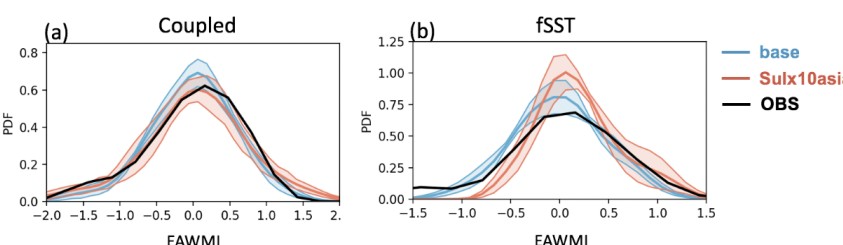

**Figure 2.** Frequency distributions of the EAWM index from (a) observations during DJF 1965-2014 (black curve)
and coupled simulations, (b) observations during DJF 1994-2005 (black curve) and fSST simulations in PDRMIP
with multimodel-means (thick coloured curves) and the associated 95% confidence intervals (coloured shades).
The confidence intervals are estimated from different models by using bootstrap resampling (e.g. Wang, 2001).









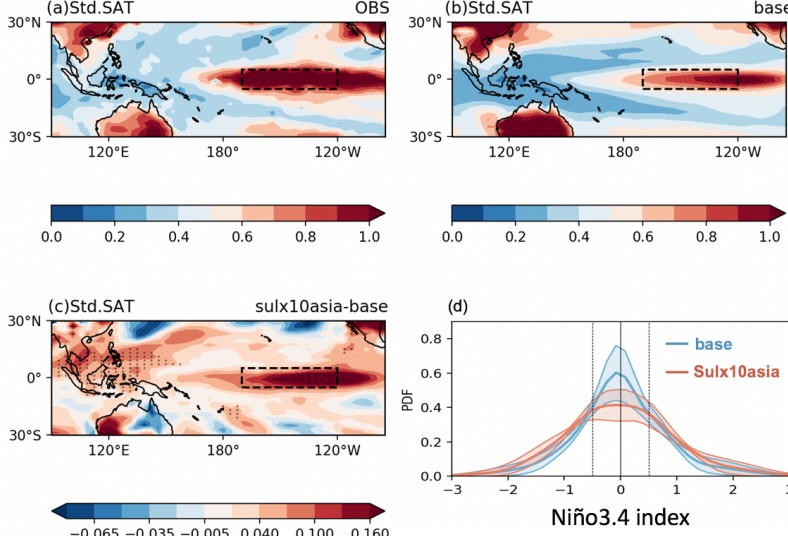

**Figure 3.** DJF multimodel mean standard deviations of SAT (SST over the ocean, ℃) from (a) observations during 1965-2014, (b) coupled baseline simulations. (c) Differences in standard deviations of SAT (SST over the ocean, ° C) between coupled SUL×10Asia and baseline simulations. Dotted regions indicate significant differences at the 95% level from the two-tailed *F*-test. (d) Frequency distributions of the Niño3.4 index from coupled simulations in PDRMIP with multimodel-means (thick coloured curves) and the associated 95% confidence intervals (coloured shades). The confidence intervals are estimated from different models by using bootstrap resampling.






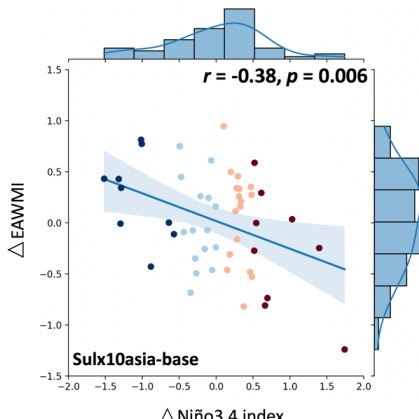

2  **Figure 4.** Joint distributions of multimodel mean differences in the EAWM index against corresponding

3  differences in the Niño3.4 index between coupled SUL×10Asia and baseline simulations, including the linear fits

4  with 95% confidence intervals.

29

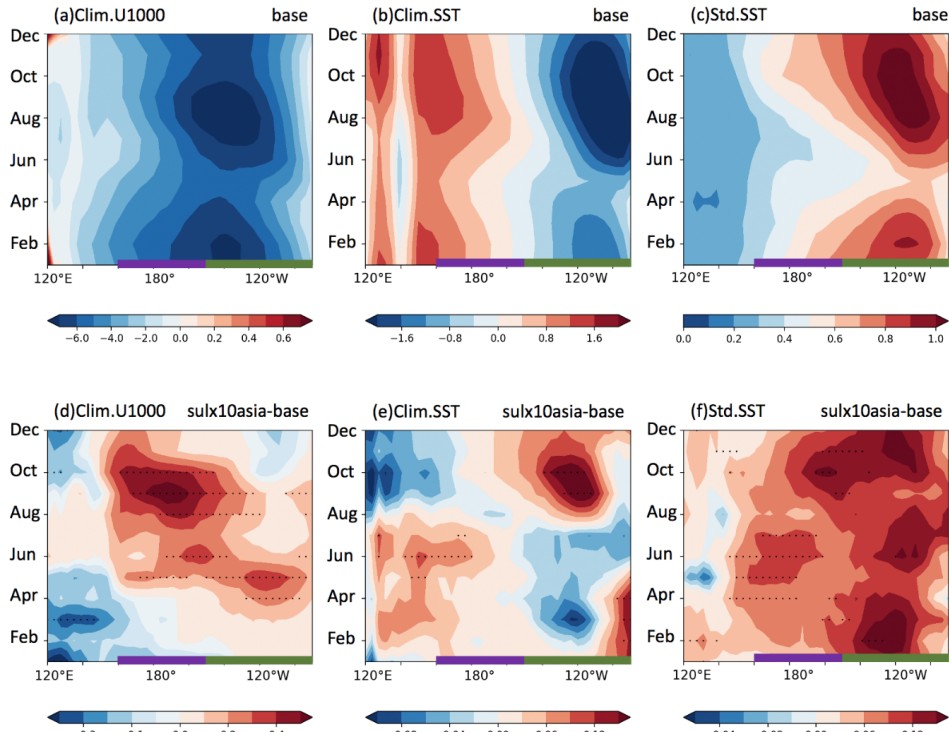

**Figure 5.** Multimodel mean longitudinal transect of the monthly climatological (a) 1000 hPa zonal wind (U1000, m s$^{-1}$), (b) SST minus zonal mean (°C), (c) SST standard deviation (°C) for the equatorial Pacific (5°S–5°N) from coupled baseline simulations; and their changes in (d) U1000, (e) SST, (f) SST standard deviation between coupled SUL×10Asia and baseline simulations. Dotted regions in (d)(e) indicate significant changes at the 95% level from the two-tailed Student's $t$ test; in (f) indicate significant changes at the 95% level from the two-tailed $F$-test. The definition longitudes of the Niño3 and Niño4 indices are marked by green and purple thick bars respectively along the x axis.



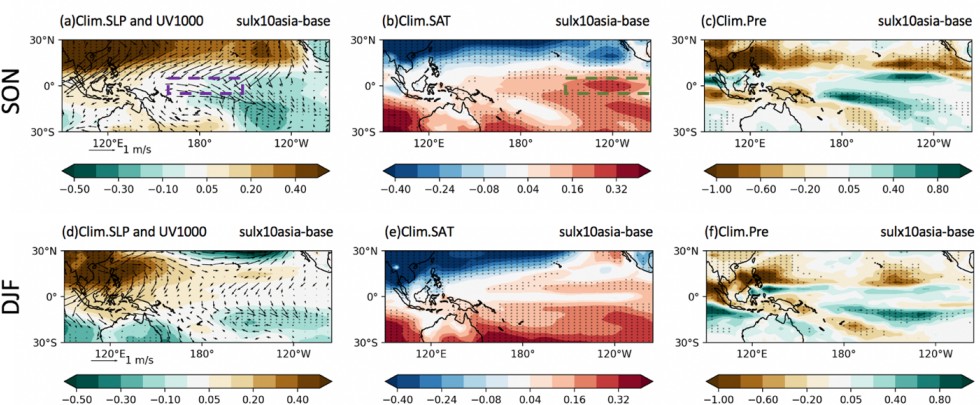

**Figure 6.** (a-c) SON, (d-f) DJF multimodel mean changes in (a)(d) sea level pressure (SLP; hPa, shading) and 1000 hPa wind (UV1000, vector), (b)(e) surface air temperature (SAT, SST over the ocean) minus domain mean (°C), (c)(f) precipitation (Pre, mm d$^{-1}$) between coupled SUL×10Asia and baseline simulations. Dotted regions indicate significant changes at the 95% level from the two-tailed Student's $t$ test. The definition regions of the Niño3 and Niño4 indices are marked by green and purple rectangles in panels a-b respectively.





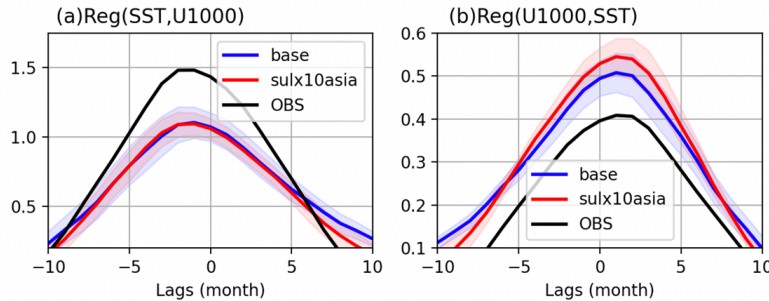

**Figure 7.** Multimodel mean lag-regression coefficients of (a) the Niño4 U1000 index onto the Niño3 SST index (indicative of the Bjerknes feedback) (m s$^{-1}$ °C$^{-1}$), (b) the Niño3 SST index onto the Niño4 U1000 index (indicative of the zonal wind forcing of SST) (°C m$^{-1}$ s) from observations (black curve) and coupled simulations in PDRMIP with multimodel-means (thick coloured curves) and the associated 95% confidence intervals (coloured shades). The confidence intervals are estimated from different models by using bootstrap resampling.



**Table 1.** Models used in this study and their specifications.

| Model | Version | Indirect effects included | References |
|---|---|---|---|
| CESM1-CAM5 | 1.1.2 | Sulfate: all indirect effects | Hurrell et al. (2013); Kay et al. (2015) |
| MIROC-SPRINTARS | 5.9.0 | Sulfate: all indirect effects | Takemura et al. (2009); Watanabe et al. (2010) |
| HadGEM3 | GA 4.0 | Sulfate: all indirect effects | Bellouin et al. (2011); Walters et al. (2014); |
| NorESM1 | NorESM1-M | Sulfate: all indirect effects | Bentsen et al. (2013); Iversen et al. (2013); |

**Table 2.** Number of El Niño and La Niña years for each model from coupled baseline and SUL×10Asia
simulations in PDRMIP.

| Years | CESM1-CAM5 (base) | CESM1-CAM5 (sulx10asia) | MIROC-SPRINTARS (base) | MIROC-SPRINTARS (sulx10asia) | HadGEM3 (base) | HadGEM3 (sulx10asia) | NorESM1 (base) | NorESM1 (sulx10asia) |
|---|---|---|---|---|---|---|---|---|
| Niño3.4 > 0.5 | 16 | 17 | 8 | 15 | 10 | 11 | 12 | 14 |
| Niño3.4 < -0.5 | 17 | 22 | 6 | 13 | 9 | 9 | 10 | 14 |

