# Peer review of "Response of the link between ENSO and the East Asian winter monsoon to Asian anthropogenic aerosols"

_EGUsphere, 2024_

## Author Comment (AC1)

We thank both reviewers for their positive comments and constructive suggestions. Below we provide our point-by-point replies to the comments (**in bold**).

**Referee: 1**

**(1) P1, L23-26: It is unclear how an improved understanding of the changes in the year-to-year EAWM variability is linked to reducing uncertainties in future projections of variability of regional extremes. There is no evidence for this claim. Suggest removing this inference.**

Previous studies have indicated that the interannual variability of the frequency and intensity of temperature and precipitation extremes during winter in the East Asian monsoon region are modulated by the intensity of the EAWM (e.g., Yang et al., 2020; Zuo et al., 2022). Although the underlying mechanisms are unexplored in this study, our findings—which highlight changes in the interannual variability of the EAWM in response to aerosol forcing—carry important implications. These results provide critical insights for reducing uncertainties in future projections of regional climate extremes, including cold surges and flooding events. To support the claim, we cite these two studies in the "Introduction" section (revised manuscript page 1, lines 34-35 and page 2, lines 1-2):

*"As such, the year-to-year variations of the EAWM have the potential to cause extreme cold disasters and severe flooding in Southeast Asian countries (e.g. Huang et al. 2012; Yang et al., 2020; Zuo et al., 2022), with consequent marked social and economic impacts (e.g. Chen et al., 2005; Zhou et al. 2011)"*

**(2) P1, L32: Feng et al (2010) is about ENSO impacts. More relevant previous studies of EAWM impacts may be added to support this statement, for example, Zhou and Wu (2010) and Wang et al. (2022) talked about the impacts of EAWM on precipitation over East Asia and western North Pacific.**

Thank you for indicating that. We have removed Feng et al (2010), and added Zhou and Wu (2010) and Wang et al. (2022) in the main text (revised manuscript page 1, lines 31-34):

*"The East Asian winter monsoon (EAWM) is one of the most prominent features of the northern hemisphere atmospheric circulation during the boreal winter and has a pronounced influence on weather and climate of the Asian-Pacific region from the northern latitudes to the equator (e.g. Chang, 2006; Zhou and Wu., 2010; Wang et al., 2022b)."*

**(3) P2, L9-11: In addition to subjecting to the ENSO impact, the EAWM variability may be partly independent of ENSO and affect climate over East Asia and the western North Pacific, as pointed out by previous studies (e.g., Wu et al. 2014; Chen et al. 2015; Wang et al. 2021, 2022), which pinpoints the importance of studying EAWM changes, such as the results of atmosphere-only model simulations in this study.**

This sentence has been rephased and these studies have been cited in the main text (revised manuscript page 2, lines 11-14):

*"The EAWM exhibits distinct interannual variability (e.g. Gong et al., 2014; Chen et al. 2015) that mainly originates from intrinsic atmospheric processes (e.g., Wu et al. 2014; Wang et al.*

*2021, 2022b) and the El Niño-Southern Oscillation (ENSO) forcing through the Pacific-East Asia teleconnection (Zhang et al., 1996)."*

**(4) P4, L10-11: What are those slow and fast responses? Please add relevant references.**

We have provided more detailed explanations of the "slow and fast responses" and added relevant references in the main text (revised manuscript page 4, lines 13-20):

*"PDRMIP offers a unique opportunity for elucidating the complexities of the ENSO-EAWM-aerosol nexus and its mechanisms, particularly with regard to the role of air-sea interactions in modulating the aerosol-driven response. Indeed, one approach that has provided valuable insights, is the decomposition of the response into two complementary components: a fast response involving atmospheric and land surface adjustments but fixed sea surface temperature (SST), acting on short timescales (a few years) and a slow response, which includes the full extent of the oceanic circulation response, thus effective on decadal or longer time scales (e.g. Samset et al., 2016; Liu et al., 2018; Dow et al., 2021; Fahrenbach et al., 2024; Liu et al., 2024)."*

**(5) P4, L15-16: A factor of 10 increase: How realistic is this 10-fold increase? Is there any evidence or previous studies for supporting this?**

The simulated step perturbation (e.g. 10-fold increase) and equilibrium response in PDRMIP differs from real-world transient adjustments but allows a clearer emergence of the signal from internal variability for sufficiently large forcing (Samset et al., 2016). This dataset has been widely used to examine the fast and slow responses of the climate system (e.g. global and regional precipitation (Liu et al., 2018); the Aleutian Low (Dow et al., 2021); Australian summer monsoon (Fahrenbach et al., 2024); East Asian summer monsoon (Liu et al., 2024)) to regional aerosol forcing. Compared to the historical emission changes, Asian sulfate aerosols increased steadily from the 1950s to the early 2000s by an overall comparable amount (e.g., Smith et al. 2011; Zhen et al., 2024,).

**(6) P4, L32: How to evaluate the performance of fSST baseline simulation as climatological SST is specified to force the model?**

In fSST baseline simulations, SSTs are fixed to the 2000 climatology. The figure below shows the DJF multimodel mean large-scale circulation patterns from (a-c) observations during 1994-2005, and (d-f) the PDRMIP fSST baseline simulations (last 12 years). This indicates that the fSST baseline simulations can well capture the winter climatological large-scale circulation over East Asia, which is dominated by the Siberian High centred over northwestern Mongolia, northerly near-surface winds along the East Asian coast, cold air over northern China and precipitation over southeastern China.

[Figure]

**(7) P6, L7-10: Disagree. According to the figures, the temperature and precipitation changes show distribution different from the anomalies in the baseline experiment. For example, the Asian land region is mostly covered by temperature decrease except for the coast. This cannot be explained using enhanced ENSO amplitude and strengthened anticyclone over the western North Pacific.**

To better compare with the baseline simulation, the results from the SUL×10Asia experiment are added to Figure 1 (panels g-I below). Taken together, panels d-l in Figure 1 show that although ENSO signal and its induced PEA pattern across the tropical Pacific enhance under increased Asian aerosols, ENSO-induced warming over southeastern and central China and cooling over northeastern China weaken. The weakened warming (i.e. cooling anomalies) over southeastern and central China appears to be primarily associated with the Asian aerosol-induced cooling (Fig. S2k), while the weakened cooling (i.e. warming anomalies) over northeastern China is related to the enhanced and northwestward-shifted PEA teleconnection pattern. The associated northwestward-expanding cyclonic anomalies (Fig. 1k) contribute to the warming anomalies over northeastern China. The comparison between changes in Niño3- and Niño4-induced circulation anomalies (new Figure S3) indicates that the latter produces a stronger warming over the Niño3.4 region and northeastern China than the former. This confirms that stronger El Niño-related warm SST anomalies over the Niño3.4 region induce a more intense PEA pattern with a northwestward expansion, featuring cyclonic anomalies and warm anomalies over northeastern China.

We have briefly mentioned this in the main text (revised manuscript page 6, lines 13-16):

*"Over land, warm anomalies over southeastern and central China weaken, as well as cold anomalies over northeastern China (Figs. 1d, g, j), primarily associated with the Asian aerosol-induced cooling (Fig. S2k) and the enhanced, northwestward-expanding PEA teleconnection pattern (Figs. 1e, h, k, Fig.S3), respectively."*

[Figure]

**Figure 1.** DJF regressions of (a)(d)(g) surface air temperature (SAT, SST over the ocean, °C, shading) and 1000 hPa meridional wind (V1000) over the broad East Asia (green contours, values plotted only when larger than 0.1 m s$^{-1}$ °C$^{-1}$), (b)(e)(h) sea level pressure (SLP; hPa, shading) and 850 hPa wind (UV850; m s$^{-1}$, vector), (c)(f)(i) precipitation (Pre, mm d$^{-1}$) onto the Niño3.4 index from (a-c) observations during 1965-2014, multimodel mean coupled (d-f) baseline and (g-i) SUL×10Asia simulations in PDRMIP. Dotted regions indicate significant correlations at the 95% level from the two-tailed Student's *t* test. Differences in regressions of (j) SAT and V1000 (green contours, values plotted only when larger than 0.05 m s$^{-1}$ °C$^{-1}$), (k) SLP and UV850, (l) Pre between coupled SUL×10Asia and baseline simulations. Dotted regions represent differences that remain significant after false discovery rate (FDR) correction of *p-values* from two-tailed Student's *t*-test (Wilks et al., 2016). The definition regions of the EAWM index and the Niño3.4 index are marked by red and black rectangles respectively.

[Figure]

**Figure S2.** Top row: DJF multimodel mean of (a) sea level pressure (SLP; hPa, shading) and 850 hPa wind (UV850; m s$^{-1}$, vector), (b) surface air temperature (SAT, SST over the ocean, °C, shading), (c) precipitation (Pre, mm d$^{-1}$) from the PDRMIP fSST baseline simulations. Second row: DJF multimodel mean changes of (d) SLP and UV850, (e) SAT and 1000 hPa meridional wind (V1000, m s$^{-1}$, green contours, values plotted only when smaller than -0.1 m s$^{-1}$), (f) Pre between SUL×10Asia and baseline simulations in the PDRMIP fSST experiments. Bottom two rows: As panels (d-f), but for the PDRMIP coupled simulations during DJF for years 50-61 (g-i), for years 50-99 (j-l). Dotted regions indicate significant changes at the 95% level from the two-tailed Student's $t$ test. The definition region of the EAWM index is marked by a red rectangle in the middle column (panels e, h and k).

[Figure]

**Figure S3.** Differences in DJF multimodel regressions of (a, c) surface air temperature (SAT, SST over the ocean, °C, shading) and 1000 hPa meridional wind (V1000) over the broad East Asia (green contours, values plotted only when larger than 0.05 m s$^{-1}$ °C$^{-1}$), (b, d) sea level pressure (SLP; hPa, shading) and 850 hPa wind (UV850; m s$^{-1}$, vector) onto the (a, b) Niño3 index, (c, d) Niño4 index between coupled SUL×10Asia and baseline simulations. Dotted regions represent differences that remain significant after false discovery rate (FDR) correction of *p-values* from two-tailed Student's *t*-test (Wilks et al., 2016). The definition regions of the EAWM index and the Niño3.4 index are marked by red and black rectangles respectively.

**(8) P6, L23-24 and L26-27: This indicates a shift in the mean state (stronger mean EAWM) due to the continent cooling induced by aerosol increase. How to understand the reduced EAWM variability under stronger mean EAWM? Any speculation?**

As shown in Figures S2d-e, Asian aerosols can lead to cooling over the emission region and the formation of an anomalous anticyclonic circulation in fSST simulations, enhancing the climatological pattern of the EAWM. In this case, strong EAWM is more likely to develop, leading to more strong-EAWM years and less weak-EAWM years (Fig. 2c). Therefore, the aerosol forcing over Asia enhances the climatological mean intensity of the EAWM and shifts its distribution toward stronger monsoon regimes, thereby decreasing interannual fluctuations in fSST simulations.

To further validate this conclusion, differences in multimodel mean standard deviations of V1000 between SUL×10Asia and baseline experiments in fSST simulations are added in Figure 2 (panel d). This shows that the prevailing northerly wind region of the EAWM, with large V1000 standard deviations (mainly along the East Asian coast) in the baseline experiment (new Fig. S4b), exhibits a decrease in SUL×10Asia simulations, further confirming the reduced EAWM variability in fSST simulations.

[Figure]

**Figure 2.** Frequency distributions of the EAWM index from observations during DJF 1994-2005 (black curve) and (a) coupled simulations during DJF for years 50-61, (c) fSST simulations during DJF for years 3-14 in PDRMIP with multimodel-means (thick coloured curves) and the associated 95% confidence intervals (coloured shades). The confidence intervals are estimated from different models by using bootstrap resampling (e.g. Wang, 2001). Differences in multimodel mean standard deviations of V1000 (m s$^{-1}$) between SUL×10Asia and baseline experiments from (b) coupled simulations during DJF for years 50-61, (d) fSST simulations during DJF for years 3-14 in PDRMIP. Dotted regions indicate significant differences at the 95% level from the two-tailed $F$-test. The definition region of the EAWM index is marked by a red rectangle.

[Figure]

**Figure S4**. (a) Frequency distributions of the EAWM index from observations during DJF 1965-2014 (black curve) and (a) coupled simulations during DJF for years 50-99 in PDRMIP with multimodel-means (thick coloured curves) and the associated 95% confidence intervals (coloured shades). The confidence intervals are estimated from different models by using bootstrap resampling (e.g. Wang, 2001). (b) DJF multimodel mean standard deviations of V1000 (m s$^{-1}$) from coupled baseline simulations. (c) Differences in DJF multimodel mean standard deviations of V1000 (m s$^{-1}$) between coupled SUL×10Asia and baseline experiments. Dotted regions indicate significant differences at the 95% level from the two-tailed $F$-test. The definition region of the EAWM index is marked by a red rectangle.

**(9) P6, L30: Southeasterly? Do you mean "southeastward"?**

Done.

**(10) P7, L20: "Consistently" > "Consistent"**

Done.

**(11) Figure 4: Need to add notations for the colors.**

Done.

**(12) P8, L15 and L19: Reduced west-east SST gradient is expected to lead to weaker advective feedback as the advection term is proportional to the mean zonal SST gradient, right?**

The advection term $(-u' \frac{\partial \bar{T}}{\partial x})$ includes both the mean zonal SST gradient and the zonal wind anomaly. Although the zonal SST gradient reduces (Fig. 5h), there are westerly wind anomalies over the central Pacific (Fig. 5g). Also, westerly wind anomalies are larger in magnitude than the decrease in the zonal temperature gradient as shown in Figure 5h, g, therefore, the advection term increases and ENSO amplitude increases (Fig. 5i). Besides, anomalous SST warming over the eastern Pacific can further strengthen westerly wind anomalies over the central Pacific (Zebiak & Cane, 1987). Previous studies have found a link between warmer SST in the eastern than in the western equatorial Pacific and an increase in ENSO amplitude (Zheng et al., 2016; Ying et al., 2019; Hayashi et al., 2020).

We have briefly mentioned this in the main text (revised manuscript page 8, lines 33-35 and page 9, 2-3):

*"Although the east-west equatorial Pacific SST gradient weakens (Fig. 5h), westerly wind anomalies over the central Pacific are larger (Fig. 5g), sustaining the eastward advection of warm water and reinforcing the positive SST anomalies over the eastern Pacific."*

*"Besides, anomalous SST warming over the eastern Pacific can further strengthen westerly wind anomalies over central Pacific (Zebiak & Cane, 1987)."*

**(13) P8, L32: remove the word "occur"**

Done.

**(14) P9, L14-15: The preceding sentences talk about the mean state changes. How they lead to the ENSO amplitude increase is explained in the next paragraph. Suggest adding "as explained below" in the end of the sentence for transition.**

Done.

**(15) P10, L9-10: Reduced west-east SST gradient is expected to be unfavorable for the advection effect. How could it lead to an increase in ENSO amplitude?**

See the reply to comment 12.

**Referee: 2**

**Major Comments:**

**(1) Clarification of Aerosol Types Considered: The manuscript focuses on the effects of sulfate aerosols over Asia, while absorbing aerosols (e.g., black carbon) are not explicitly discussed. Previous studies have shown that absorbing aerosols can have comparable influences on atmospheric circulation and precipitation, despite their lower atmospheric burden. In particular, Asian absorbing aerosols are known to significantly affect the Walker circulation, a key process examined in this study. While it is valid to focus primarily on scattering aerosol effects, this should be clearly specified in the title and abstract to avoid concerns about overlooking absorbing aerosol impacts.**

Done. We have specified that this study focuses on the effects of sulfate aerosols over Asia in the title and abstract.

**(2) Lack of Direct Analysis of Aerosol Perturbation and Effects: While the study systematically investigates the climate responses to aerosol perturbations, it does not directly show the aerosol changes and effects themselves. Adding patterns and seasonality of aerosol optical depth (AOD) and effective radiative forcing (ERF) would provide a more intuitive representations of the aerosol perturbations, which could also help explain the spatial and seasonal climate responses.**

The sulfate loading and ERF (calculated using the method of Richardson et al., 2019) patterns in autumn (SON) and winter (DJF) have been added as new Figure S7. This shows that sulfate loading mainly concentrates over East Asia and South Asia in the baseline simulations for both seasons (Fig. S7a, d), with a larger magnitude in SON when crop residue burns. In SUL×10Asia simulations, sulfate loading increases approximately tenfold (Fig. S7b, e), generating negative ERF anomalies over Asia and surrounding oceans (Fig. S7c, f). In SON, midlatitude westerlies (Fig. S6a) transport aerosols downstream, extending the region of negative ERF anomalies to the northwestern Pacific. In DJF, the Siberian High strengthens, and the Aleutian Low deepens with a southward shift compared to SON (Fig. S6d). The associated northwesterlies along the eastern flank of the Siberian High and northeasterlies along the coast of East Asia confine the negative ERF anomalies primarily south of 30°N. The difference in spatial patterns of ERF anomalies between these two seasons broadly aligns with the difference in the anticyclonic anomalies in Figure 6a, d, although ERF-induced radiative cooling will further reduce sea surface temperatures in coupled simulations (Fig. 6b, e), thereby amplifying initial circulation anomalies through air-sea coupling.

We have briefly mentioned this in the main text (revised manuscript page 9, lines 14-17 and lines 25-27):

*"The associated midlatitude westerlies (Fig. S6a) transport aerosols downstream, extending the region of aerosol-induced negative effective radiative forcing (ERF) anomalies to the northwestern Pacific (Fig. S7a-c), leading to the zonally wider cooling."*

*"The associated northwesterlies along the eastern flank of the Siberian High and northeasterlies along the coast of East Asia confine the negative ERF anomalies primarily south of 30°N (Fig. S7d-f)."*

[Figure]

**Figure S7.** (a) SON, (d) DJF multimodel mean sulfate loading ($10^{-7}$kg/m$^2$) from the PDRMIP fSST baseline simulations. Changes in (b, f) sulfate loading and (c, e) effective radiative forcing (ERF, W/m$^2$) between fSST SUL×10Asia and baseline simulations in these two seasons. Dotted regions indicate significant changes at the 95% level from the two-tailed Student's *t* test.

[Figure]

**Figure S6.** As in Figures S4a-c, but from coupled baseline simulations for (a-c) SON, (d-f) DJF.

[Figure]

**Figure 6.** (a-c) SON, (d-f) DJF multimodel mean changes in (a)(d) sea level pressure (SLP; hPa, shading) and 1000 hPa wind (UV1000, vector), (b)(e) surface air temperature (SAT, SST over the ocean) minus domain mean (°C), (c)(f) precipitation (Pre, mm d$^{-1}$) between coupled SUL×10Asia and baseline simulations. Dotted regions indicate significant changes at the 95% level from the two-tailed Student's $t$ test. The definition regions of the Niño3 and Niño4 indices are marked by green and purple rectangles in panels a-b respectively.

**Specific Comments:**

**(1) The analysis uses different time periods for the coupled simulations (1965–2014) and fixed-SST simulations (1994–2005). Please explain the reasoning behind this difference. Additionally, please double-check the results to ensure that the different periods do not introduce inconsistencies in the comparison.**

Thank you for indicating this. For consistency, we now select 12 winters from both coupled (DJF for years 50-61, after the initial spin up) and fSST (DJF for years 3-14) simulations to compare with observations during DJF 1994-2005 in Figures 2a, c (provided in our response to Reviewer 1, Comment 8). In line with our previous findings, in coupled simulations the multi-model mean EAWM amplitude increases by 18% due to the Asian aerosols, together with more extreme EAWM years in the SUL×10Asia experiment. In fSST simulations, the multi-model mean EAWM amplitude decreases by 19%, accompanied by more strong-EAWM years and fewer weak-EAWM years in SUL×10Asia experiments. Note that specific changes in the tails of the distributions in coupled simulations may differ when using different periods (i.e. 12 and 50 winters) because of sampling. We also add differences in multi-model mean standard deviations of V1000 between coupled SUL×10Asia and baseline experiments in both coupled (both 12 and 50 winters) and fSST (12 winters) simulations in Figure 2 (panels b, d) and Figure S4 (panel c; provided in our response to Reviewer 1, Comment 8), which further confirm the opposite variations in the interannual variability of the EAWM to Asian aerosols in fully coupled and atmosphere-only experiments. These results indicate that 12 winters from coupled simulations (DJF for years 50-61) share consistent results as 50 winters (DJF for years 50-99), suggesting that the findings are not significantly sensitive to the length of the analysis period.

**(2) In PDRMIP, aerosol perturbations cover the entire Asian region (10°–50°N, 60°–140°E), but most figures cover only East Asia. This could lead to an overlook of South Asian aerosol effects. It would be helpful to demonstrate the AOD and ERF of SAsia aerosols and discuss the potential contributions.**

Added sulfate loading and ERF in Figure S7 and the reproduced climatological circulation patterns in Figures S2, S6 all show that increased sulfate loading, reduced ERF, cooling and associated anticyclonic anomalies in SUL×10Asia simulations are consistent over both East Asia and South Asia. In this case, the conclusions regarding the impact of Asian aerosols in our study inherently reflect the combined effects of both East Asian and South Asian aerosols.

**(3) The results from the SUL×10Asia experiment should be added to Figures 1, 3, and 5, to better compare with the baseline simulation.**

Done.

**(4) Statistical Significance Testing: Please include statistical significance tests for the SUL×10Asia – baseline results to ensure the statistical significance of the climate responses. And for the significance test, the method by Wilks (2016, https://doi.org/10.1175/BAMS-D-15-00267.1) will be more robust than the Student's t-test.**

Done. We have included statistical significance tests for the SUL×10Asia – baseline results with FDR correction introduced by Wilks et al., (2016) in Figures 1j-l (provided in our response to Reviewer 1, Comment 7).

**(5) Please add units in the plots for easier reading.**

Done.

**(6) The authors need to double-check the Supplementary figures:**

- **Figure S4 captions: "As in Figures 1a-c, but for SON", but variables in Fig1 and S4 are different, please double check.**
- **P9 L9 *"Aleutian Low deepens with a southward shift in the coupled baseline simulation (Figs. S4a, S2a)."* While Fig.S2a shows fixed-SST SLP and 850hPa UV**

Done.

**References**

[revised manuscript text omitted]

---

## Author Response (AR2)

We thank both reviewers for their positive comments and constructive suggestions. Below we provide our point-by-point replies to the comments (**in bold**).

**Referee: 2**

**Minor comment:**

**1. Please include the description of the method used to calculate the effective radiative forcing (ERF) in the Data and Methodology section. Since different approaches can lead to varying results (Tang et al., 2019), this information is essential for clarity and reproducibility.**

**Tang, T., Shindell, D., Faluvegi, G., Myhre, G., Olivié, D., Voulgarakis, A., ... & Smith, C. (2019). Comparison of effective radiative forcing calculations using multiple methods,**

Thank you for indicating this. We have added one sentence clarifying the calculation method in the main text (revised manuscript page 5, lines 2-4):

*"The effective radiative forcing (ERF) is calculated as the difference in the top of the atmosphere net radiative flux between the SUL×10Asia and baseline fSST simulations (Samset et al. 2016)."*